# Dual Modular Titanium Alloy Femoral Stem Failure Mechanisms and Suggested Clinical Approaches

**DOI:** 10.3390/ma14113078

**Published:** 2021-06-04

**Authors:** Jan Zajc, Andrej Moličnik, Samo Karl Fokter

**Affiliations:** Department of Orthopaedic Surgery, University Medical Centre Maribor, 5 Ljubljanska Street, 2000 Maribor, Slovenia; andrej.molicnik@ukc-mb.si (A.M.); samo.fokter@guest.arnes.si (S.K.F.)

**Keywords:** total hip arthroplasty, titanium, prosthesis failure, dual modularity, corrosion

## Abstract

Titanium (Ti) alloys have been proven to be one of the most suitable materials for orthopaedic implants. Dual modular stems have been introduced to primary total hip arthroplasty (THA) to enable better control of the femoral offset, leg length, and hip stability. This systematic review highlights information acquired for dual modular Ti stem complications published in the last 12 years and offers a conclusive discussion of the gathered knowledge. Articles referring to dual modular stem usage, survivorship, and complications in English were searched from 2009 to the present day. A qualitative synthesis of literature was carried out, excluding articles referring solely to other types of junctions or problems with cobalt-chromium alloys in detail. In total, 515 records were identified through database searching and 78 journal articles or conference proceedings were found. The reasons for a modular neck fracture of a Ti alloy are multifactorial. Even though dual modular stems have not shown any clinical benefits for patients and have been associated with worse results regarding durability than monolithic stems, some designs are still marketed worldwide. Orthopaedic surgeons should use Ti6Al4V dual modular stem designs for primary THA in special cases only.

## 1. Introduction

Monoblock femoral stems have traditionally been used in total hip arthroplasty (THA) to treat patients with progressive joint wear. Soon, modular heads with different bore lengths and material properties will be added to most commercially available THA systems. The head and the stem are joined with a circular cross-section Morse taper intraoperatively. In the last 30 years, implants with dual modular stems consisting of a modular (interchangeable) neck that is separate from both the stem and the head have been introduced to improve biomechanical restoration. The three parts of this dual modular stems fit together via an additional tapered joint, i.e., the neck-stem taper. For the purpose of this manuscript, the term “monoblock” will be used to refer to an implant with a monolithic stem coupled with a standard modular head (Figure 1).

Several neck designs allow the surgeon to optimize leg length, centre of rotation, and hip stability during surgery [1]. Re-establishing appropriate hip biomechanics can contribute to a slower wear of THA components and may ease the revision of acetabular components by temporarily removing the interchangeable neck [2].

Corrosion at the neck-stem junction has been a particular concern because modular neck fractures usually occur at this level. The modular neck and the body are typically joined by a rectangular cross-section taper with rounded fillets. One of the first commercially available dual modular cementless femoral stems for primary THA was the GSP implant (Cremascoli S.p.A., Milan, Italy), where the body and the neck part were made of a Ti6Al4V alloy (ASTM F136). The neck had a 12/14 cylindrical taper at the proximal end and a 9 × 18 mm rectangular taper at the distal end. The company modified the design of the body a few years later and the system was named An.C.A. Fit. Later, the Italian company was bought by Wright Medical Technology (now MicroPort) Inc., Arlington, TN, USA. The body was further changed to the Profemur series, i.e., cementless, straight Ti6Al4V alloy-made dual modular stems with a grit-blasted surface finish. Despite the evolution of the body part, no significant design changes were introduced in the neck-body taper joint and the system was sold worldwide (Figure 2). Soon, similar dual modular systems for primary THA appeared in the portfolios of most orthopaedic implant producers. As the popularity of dual modular systems continued to grow [3], unpredicted complications due to increased modularity were reported. Implant fractures, corrosion and metallosis have been the main culprits behind a general two-fold increase in revision rates [4]. Despite these failures having a common denominator, they can be, for practical purposes, divided into mechanical and inflammatory complications of modularity and will be discussed separately.

Concern exists about the corrosion of tapered connections in biological tissues, especially at the neck-stem junction. After some implants were removed from the market because of a high failure rate, the enthusiasm of orthopaedic surgeons for dual modular stems has vanished [5]. This narrative review aims to describe the currently known facts about the mechanisms of failure, risk factors, and the number of modular neck fractures based on peer review articles to provide knowledge on how to manage patients after a primary THA with implanted dual modular femoral stems.

## 2. Materials and Methods

Following the Preferred Reporting Items for Systematic Reviews and Meta-Analyses (PRISMA) [6] guidelines, we have systematically reviewed the literature from 2009 onward to identify studies that have evaluated dual modular stem prosthesis survivorship, complications with clinical use, mechanical complications, and biological responses. We have searched through the National Center for Biotechnology Information (NCBI) database (Pubmed) on 1 January 2021, and through the ScienceDirect and Google Scholar databases on 1 April 2021.

The following Medical Subject Headings (MeSH) terms were used to search in the NCBI (PubMed) database: “arthroplasty, replacement, hip”, “prostheses and implants”, “hip prosthesis”, “prosthesis failure”, “corrosion”, “adverse effect”, and “long-term adverse effects”. Studies on animals were excluded. We used the following search terms to search in ScienceDirect and Google Scholar databases: “corrosion”, “total hip arthroplasty”, “adverse effect”, and “dual modular.”

## 3. Results

In total, 515 records were identified through the database search. After duplicates were removed and records were screened for relevance, 78 articles were included in the qualitative synthesis.

We have excluded 437 studies related to head-neck trunnion complications, stem-sleeve modularity, specific details about cobalt-chromium (CoCr) junctions, or those not otherwise connected to our topic.

Titanium (Ti) alloy (Ti4Al6V) modular neck complications are described in more detail below. Problems with CoCr interchangeable necks are also mentioned, since CoCr necks were meant to solve the problems with Ti alloy necks (Appendix A).

### 3.1. Mechanical Complication

The first clinical results with dual modular stems were promising. A study of the An.C.A. Fit (Cremascoli S.p.A.) implant found no clinical or radiographical complications due to modularity when observing patients from 5 to 10 years after their primary THA [7]; however, in 2009, Kop et al. reported that even with modern taper designs and the use of corrosion-resistant materials, increased modularity could cause metal ion generation, crevice and fretting corrosion, and the production of particulate debris, which may contribute to periprosthetic osteolysis and loosening [8]. From 2010 onwards, there is an increasing amount of scholarly articles reporting dual modular stem design prosthesis failure, starting in the form of case reports, where sudden fractures of the modular neck at the neck-stem junction were reported as early as two years after implantation [9,10,11,12]. Atwood et al. reported on fractures of modular titanium (Ti) alloy necks coupled with Ti alloy stems. Fretting at a modular junction continually wears away the passive oxide layer that provides Ti6Al4V alloy implants with protection from corrosion, requiring constant repassivation. Repassivation in a crevice depletes the limited oxygen supply, thus decreasing the local pH and promoting further corrosion, resulting in pitting and the formation of sharp cracks [9]. These fatigue cracks typically occur on the lateral proximal side of the male part of the oval-shaped Morse taper joint, where larger tensile stresses appear during activities of daily living.

Thus, to minimize micromotions, Ti alloy necks were replaced with CoCr necks due to the modulus of elasticity of a CoCr alloy being twice as high as that of a Ti6Al4V alloy [13]; however, Gilbert et al. also noted cracks on the medial proximal side of a modular neck in the neck-stem taper area [14]. Using backscattered electron microscopy, the authors demonstrated that the crack propagation process is one of corrosion cracking, where the crack tip stresses arise from the oxide formation in the crack and not from externally applied tensile stresses [14]. In CoCr necks, many crack nucleation sites are found to be randomly positioned around the perimeter of a fracture surface [15]. The intergranulated corroded areas likely act as local stressors under fatigue loads [15]. Fatigue, fracture, corrosion, dissociation, subsidence, and loosening are potentially problematic mechanical complications [16]. CoCr necks show better fretting fatigue resistance than Ti4Al6V, but the exact opposite concerning fretting and wear resistance [17].

After the case reports mentioned above, a handful of single-surgeon and single-centre studies reporting experiences with dual modular stems were published. One such study reporting modular neck fractures in six patients with 5.6 years mean time to revision found the use of a long neck and high BMI to be related in some of the causes of increased bending stresses [18]. Pour et al. reported a 6% modular neck fracture rate at a mean follow-up of 4.2 years [19]; however, long-term studies of original implants used on the north Italian population with a low BMI showed a low modular neck fracture rate [20]. The Italian Arthroplasty Registry Project’s fourth report shows a 0.26% fracture rate for modular necks as a cause for revision surgery [21]. Finally, a nationwide study of the Profemur Z (Wright Medical Technology Inc.) dual modular stem, implanted in 2767 patients, has shown that modular neck fracture occurred in 0.83% of observed patients, with a mean BMI of 29 kg/m^2^ [22]. Fractures occurred more commonly in the group with modular necks made from a CoCr alloy (1.44%) than with those with Ti alloy necks (0.76%) [22]. An annual report of the German Arthroplasty Registry presents a decrease of over 50% in the usage of different types of stem modularity in primary THA from 2017 to 2019 [23].

A retrieval study of 134 heads, 60 stems, and 8 different bearing designs from 5 manufacturers was performed to evaluate fretting and corrosion damage in different metal-on-metal (MoM) bearings [24]. Evidence of taper damage was found in all types of examined modular interfaces and all distal modular neck tapers were affected. Higher damage scores at the head-stem interface were observed when heads were coupled with a modular neck. Larger head sizes, increased medio-lateral offsets, longer neck moment arms, and dissimilar alloy pairings were associated with increased taper damage in modular components [24]. A retrospective single surgeon cohort study of 152 hips (Profemur Z, Wright Medical Technology Inc.) showed severe neck-body taper corrosion in 2/3 of patients revised for hip pain [25]. Overall, the Kaplan–Meier survival rate showed an 8.6% revision rate for stem-specific causes of failure [25]. Extensive corrosion at the tapered neck-stem junction was revealed 2.5 years after undergoing a primary THA [26]. Researchers agreed that it was necessary to better understand the nature of the wear and corrosion at the neck-stem taper interface [27,28].

Male sex, longer lever arm, higher BMI, younger age, longer time since implantation, and an active lifestyle are all known factors for higher complication risk [11,18,19,22,29,30,31,32,33].

A study of 60 cases of Rejuvenate (Stryker, Mahwah, NJ, USA) stem implantation showed evidence of severe neck-stem corrosion in all cases, but no corrosion on head-neck tapers [34]. In total, 5% of implants were revised less than 4 weeks after index surgery and all showed fretting and corrosion, confirming that the destructive processes start early on and that the rates increase over time [34]. A retrospective study from 2018 reports a 34% revision rate of Rejuvenate (Stryker) dual modular stems with a mean time to revision of 2.7 years [35]. The largest areas of material loss with Rejuvenate (Stryker) implants were proximally located at the medial conical part of the neck taper [36]. Another study of 73 hips at a mean follow-up of 4.2 ± 0.6 years showed an 86% clinical failure rate and 78% of hips required revision at a mean of 3.2 ± 1.0 years [5]. By grading fretting and corrosion with Goldberg’s criteria [37] on a scale from one (none) to four (severe) and comparing the results with the duration or length of implantation (LOI) shows an interesting pattern [34]. There is a positive correlation between the LOI and corrosion score and a negative correlation between the LOI and fretting score. The authors explain the latter with more and more of the surface becoming obscured over time by increasingly large amounts of adherent corrosion debris. One of the major causes of fretting and corrosion is the process of cyclic cantilever bending [34].

Two dual modular femoral stem designs ABG II and Rejuvenate (Stryker) were voluntarily recalled and Profemur Z (Wright Medical Technology Inc.) was redesigned and remains under scrutiny due to extremely poor clinical records [2]; however, evidence of neck-stem taper corrosion was not only observed in “infamous” models known for being recalled from the market, as clearly shown by a report from a patient with progressive hip pain five years after implantation [38]. The authors report corrosion at the neck-stem taper previously not described in M/L Taper Kinectiv (Zimmer Biomet, Warshaw, IN, USA) implant [38]. A comparison of corrosion and fretting behaviour between recalled designs and modular neck designs still in use has revealed that corrosion and fretting have occurred on all observed modular neck-stem implants regardless of design, but corrosion has been worse for mixed-metal couples than homogenous couples [39].

Variability in outcome results between different patients with CoCr implants can also be attributed to the broad variability in implant microstructures [22,40]. Because implant alloy microstructures are not sufficiently standardized, they varies significantly for both cast and wrought alloys, not only between manufacturers, but also within the same implant design [41]. The most severe corrosion damage is exhibited for alloys with carbides due to phase boundary corrosion, hard phase detachment, and consequent metal ion release [41]. A combination of an extra-long head and a long interchangeable Ti alloy neck in a dual modular stem prosthesis can, in certain biomechanical situations, such as stumbling, when forces greater than eight times the patient’s body weight may apply, create tensile stresses so high that they surpass the limits of the tensile strength of a Ti6Al4V alloy [42,43,44,45]. A recent retrospective study of 50 THAs with Profemur Z (Wright Medical Technology Inc., dual modular) and 52 THAs with Alloclassic–Zweymüller (Zimmer, monoblock) of otherwise similar design performed by a single surgeon demonstrated, on average, higher magnitudes of shear forces appearing with dual modular femoral implants compared to monoblock stems [46].

### 3.2. Inflammatory Complications

The effects of increased neck-stem modularity on the incidence of metal hypersensitivity reactions following THA were still unknown in 2012 [47]. Hsu et al. reported on a a patient who presented one year after a primary THA with seven months of increasing hip pain and disability due to a pseudotumor formation resulting from neck-stem junction corrosion in an otherwise well-fixed stem [47]. Patients have been presenting with progressive hip pain exacerbated while sitting as early as six months after a primary THA [48]. Two out of three retrieved stems from revisions due to pseudotumor formation causing post-operative pain in a series of 35 patients showed evidence of corrosion at the neck-stem taper [49]. Accelerated corrosion at the neck-stem interface has led to an early-onset adverse inflammatory tissue reaction (ALTR) presented with progressive swelling and pain [48].

Corrosion at the neck-body junction between the Ti alloy stem and the CoCr neck can lead to a release of debris and ions, resulting in large soft tissue masses with damaged surrounding tissues [50]. Patients with dual modular stems display neo-synovial proliferation and necrosis with an infiltrate rich in GATA-3 transcription factor and CD4 immunoglobulin. In this case, both are indicators of increased CD4 effector cells (T-helper cells) differentiation [51]. Inflammatory cells mixed with scattered metallic debris are found as focal cellular areas inside a larger field of necrosis [52]. All radiographically observed patients with Rejuvenate (Stryker) dual modular stems showed synovitis and 85% showed synovial decompression [53]. The adoption of Co alloy necks increases the risk of ALTR due to fretting and corrosion products spreading [54], resulting in a need for revision as early as one year after the implantation due to osteolysis, aseptic lymphocyte-dominated vasculitis-associated lesions (ALVALs) with a large pseudotumor, and skin hypersensitivity reactions [55]. ALTR caused a 20% incidence of revision in a cohort of 44 Rejuvenate (Stryker) implants with a 27% complication rate [56].

Even though the known factors for an overall increased complication risk (as described above in the mechanical complications section) contain male sex and high offsets, one multivariate logistic regression analysis of 118 THAs in 107 patients showed that a low neck offset (132°) and female sex presented a higher risk for failure due to an ALTR resulting from corrosion [57].

Although pseudotumors in a metal-on-metal (MoM) bearing were already a known issue in 2014, Messana et al. reported on the case of pseudotumor formation due to corrosion at the modular junction with a ceramic-on-polyethylene acetabular bearing resulting in revision surgery [58]. In the same year, Walsh et al. reported on intra-operative findings for 103 THA cases [59]. A high rate of soft tissue destruction and corrosion was present, resulting in the production of corrosive sludge and necrotic rinds and pseudotumor formation in well-fixed stems. During revision, an extended trochanteric osteotomy (ETO) was frequently required, despite advanced stem extraction techniques, leading to increased morbidity. All revised stems showed corrosion debris at the neck-stem junction [59]. There will be more complications and a higher reoperation rate if a patient has a large amount of tissue damage, potentially leading to poor treatment results [60].

Nano-analyses of wear particles showed distinctively different distributions and shapes of particles when comparing three major classes of hip implants associated with an ALTR, namely, MoM hip resurfacing arthroplasties, MoM large head THAs, and non-MoM dual modular stem THAs [61]. Particles from a dual modular stem THA are bigger and more needle-shaped when compared to the other two groups, leading to the group of patients with dual modular implants having the most severe reactions of all three groups. Ti and vanadium (V) particles have been mixed with Cr, Co, and molybdenum (Mo) particles in patients with a dual modular design but were separated from them in the other two with an ALTR in the commonly associated groups [61]. A tribocorrosion-associated ALTR also leads to a unique tissue gene expression profile that differs from the one associated with non-metallic implant wear products [62]. Co and Cr ions suppress DNA repair pathways, causing defective gene expression [2]. The long-term sequelae of these alterations are cancers that can even occur at concentrations considered as sub-toxic. Co ions can prompt greater macrophage and lymphocyte death by stimulating tumour necrosis factor-alpha (TNF-α) secretion and macrophage apoptosis [2]. A similar complication was also known with Ti alloy implants, as is evident from the case report of an osteogenic osteosarcoma formed 10 years after the implantation of an An.C.A.-Fit (Cremascoli S.p.A.) dual modular femoral implant [63]. The morphological mechanisms of evidence observed for the histology cannot be solely attributed to an elevation in metal ion levels and a multifactorial display of complex interplay between patient and implant factors is most likely the case [64]. Complications related to using dual modular stems (Rejuvenate, Stryker) outweigh their potential benefits [57].

### 3.3. Clinical Approach

Patients with an ALTR present with pain or poor hip stability [65]. It has become clear that there should be a standardized method to assess for an ALTR in a patient. There should be a low threshold to systematically evaluate patients with a dual modular stem THA, since early recognition and diagnosis will facilitate the initiation of appropriate treatment [66]. In patients with unexplained pain that have dual modular implants where the neck is made from CoCr, one may consider an ALTR as a product of mechanically assisted crevice corrosion (MACC) of the neck-stem interface [67]. A two-year follow-up of 195 hip implants in 183 patients with only moderate elevation in serum Co levels showed abnormal imaging studies in 36%, clinical symptoms in 44%, and revision due to corrosion in 13% of observed hips [68]; however, at a midterm mean follow-up of 5 years, the prevalence of revision surgery due to corrosion almost doubled [69]. Early short-term outcomes of the Rejuvenate (Stryker) implant show that the majority (66%) of patients experiencing complications require revision surgery at the end [70].

It was shown in a cohort of 198 revisions that the extent of intraoperatively found tissue damage due to an ALTR significantly correlates with dislocation and revision rates, also predisposing patients to a 20% complication rate and 8% re-revision rate [71]. Although ALTRs were commonly associated with progressive pain, Kwon et al. pointed out that the absence of symptoms does not exclude the presence of an ALTR in patients with implanted dual modular stems [72]. Elmallah et al. stated that patients could expect an improvement in pain and functioning after revision but an overall worsening in mental component scores (SF-12) [73].

Guidelines developed to evaluating MoM bearings in THA should not be adopted without caution in the context of dual modular stems, as the generation of metal ions may also occur via different mechanisms [27]. A systematic evaluation of a patient with dual modular stem should include a focused clinical history, detailed clinical evaluation, inflammatory markers, metal ion levels, hip joint aspiration tests, radiograph data, and cross-sectional imaging [74,75]. Serum Co levels >2.8 μg/L and a Co/Cr ratio >3.8 are useful clinical adjuncts in the systematic evaluation of a corrosion-related ALTR, but they should not be relied on as the sole diagnostic parameters [76]. Retrospective data presented in 2020 showed elevated Co and Cr levels in only 55% of patients with a pseudotumor [77]. One study also presented no notable correlation between serum Co and Cr levels and pseudotumor formation or size; however, it did find a correlation between Ti-levels and the size of the pseudotumor [78]. Such inconclusive metal ion levels correlations support the use of routine follow-up metal artifact reduction sequence magnetic resonance imaging (MARS MRI) for patients with dual modular CoCr stems [74,77]. In symptomatic patients with low blood ion levels, intraarticular levels may be useful in confirming MACC since they are 100 times higher than blood levels in affected patients [79].

Taper corrosion originating from a modular neck is typically most successfully treated with revision [80]. The removal of well-fixed dual taper femoral components presents a challenge [81] and the additional complication of failure to disassemble the implant further increases the complexity and morbidity of the revision surgery [4] (Figure 3). Advanced femoral revision techniques (e.g., dual chisel technique and top-out technique) should be used to remove the dual modular stem, but the surgeon should have a low threshold to perform a femoral osteotomy with stem debonding or an ETO if the removal of the implant proves difficult otherwise [3]. Kwon et al. presented a “top-out” technique that preserves the bony envelope as an advanced alternative to an ETO [81]. In one comparison of monoblock and dual modular stems, all dual modular stem revisions required extraction due to severely damaged neck/stem junctions, suggesting that it would be best to eliminate the use of dual modular stems with large-diameter heads until the design of neck-stem junctions is significantly improved [82]. In the weeks following the removal of dual modular implants from patients, a statistically significant and clinically relevant drop in serum Co, Cr, erythrocyte sedimentation rate (ESR), and C-reactive protein (CRP) levels can be observed [83].

## 4. Discussion

It is very hard to predict the lifespan of a particular implant. Studies concerning dual modular implants are not unanimous regarding implant survival rate. A nationwide study of 2767 hips has shown a time to fracture of 3.4 years (SD ± 1.4) for implants with a CoCr neck and 5 years (SD ± 2.3) for implants with Ti alloy necks [22]. Even taking all known predisposing factors into account, it currently appears impossible to accurately pinpoint when a fracture is going to occur. A recent report for a patient with the same type of dual modular stem implanted on both hips described a modular neck fracture after 3 years on the left side and after 20 years on the right side post-implantation, respectively, with identical risk factors applying for both implants [43].

Even though the mechanical loading of an implant can lead to crack initiation and propagation, it is not necessary for corrosion to take effect. The submersion of test bodies from four different frequently used alloys in serum has shown that 80% of the entire dissolution of ions happens in the first 24 h, but, more importantly, it happens without mechanical influences [84]. When a crack is already formed past a certain point, there is no need for mechanical load for its propagation. Self-propagating autocatalytic processes occur that, selectively dissolve the beta phase of the alloy. A loss of alpha grains and their subsequent conversion into oxide follows [14].

Acetabular cup deformation, an inevitable mechanical consequence in uncemented press-fit components, can lead to a breakdown in lubrication mechanisms resulting in suboptimal conditions [85]. Furthermore, a large, thin-walled acetabular cup may be deformed by the deforming periacetabular bone when the hip is loaded. Consequently, it squeezes the large-diameter prosthetic heads, resulting in an increased frictional moment. The projection of the frictional moment onto the neck axis produces a torque on the neck which is transferred within the taper junctions, leading to greater corrosion-promoting micromotion [86]. Binary logistic regression has shown that each 4-mm increase in head diameter increases the chance of neck fracture by a multiplicative factor of 3.21 for the Profemur Z and Profemur E hip systems [19].

To minimize the impact of corrosion, we need to further explore the prevention of micromotion, the role of implant alloy metallurgy in the corrosion process, the in vivo generation of corrosive environment, and potential unanticipated problems [87]. To protect alloys from corrosive processes, different coatings have shown some promise. A chitosan/diclofenac coating originally invented to achieve a more controlled and site-specific administration of a non-steroidal anti-inflammatory drug (NSAID) has shown an improvement in corrosion resistance for AISI 316LVM stainless steel [88]. To achieve good long-term clinical outcomes, coatings should be thick enough to withstand the high contact stresses and they should avoid delamination [89]; however, some coatings can be worn off to a macroscopically apparent degree with relative ease [90], thus preventing their use on taper junctions. Currently, we are in most cases still relying on specific oxide-forming alloy properties in terms of uniform corrosion attack prevention.

Stability at the neck-stem junction can be significantly reduced if impaction during the assembly is not directed in line with the longitudinal axis of the taper junction, as occurs with 8° and 15° necks [91]. Taper cleanliness also plays a very big role in preventing micromotion, as a comparative study has shown. In in vitro conditions, artificially contaminated taper junctions with porcine bone chips have created micromotions that are more than twice as large than those of clean taper junctions [92]. Some relative motion at the neck-stem junction occurs due to the shape of the tapers deviating from an ideal design dimension [93]. Three-dimensional contact colour maps clearly indicate that the surfaces are not in contact over the entire circumstances but typically at four localized lines of contact. Some implants also show one surface in contact or at closer proximity than the contralateral surface. The cause of the deviation in dimensions is most probably machine tool deflexion during machining [93].

More rigorous and realistic preclinical testing should be done before new types of modular junctions are introduced in clinical practice [94]. When surgeons start to use a new implant design, there should be a 5-year follow-up or, even better, 5-year post-market surveillance studies before introducing newer alternate bearings and/or junctions in THA on a widespread scale [95]. Simulations of fretting corrosion under different conditions have shown that preclinical protocols should adopt testing implant in bovine calf serum and some studies have also suggested simultaneous testing at different pH levels [96]. Immersion in bovine calf serum instead of Ringer’s solution significantly decreases the endurance limit of the taper junction [97]. Our search method identified no studies that would expose the implant and the solution to an electric current simulating the galvanic processes as they occur in the body.

Taper corrosion leading to an ALTR can mimic other more common diagnoses, like instability, aseptic loosening, or periprosthetic joint infection [98]. Serum metal ion levels are currently the best screening tool for corrosion [98]; however, they should never be used as a sole investigative tool, but rather as a part of a more complete clinical and radiographic evaluation [99]. Furthermore, they do not correspond with the other factors for modular neck fracture very well, making them an inefficient tool in screening patients with dual modular implants for upcoming modular neck fracture, especially in TiTi alloy neck-stem combination [100]. In their study, the authors pointed out that the highest Ti, V, and Al serum levels were measured in a hard-working, traditional farmer who was a part of low-risk group (straight, short neck; lower BMI) [100]. This is similar to the observed changes in Co serum level dynamics when responding to activity intensity modification as reported two years earlier [101]. Pre-revision metal ion levels are also not correlated to the amount of intraoperatively found tissue damage [64]. Further studies are needed to identify patients with threatening interchangeable neck fracture after primary dual modular stem THA [100].

A comparison of 284 patients with monoblock stems and 594 patients with dual modular stems showed no difference in clinical outcome scores [102]. Thus, dual modular stems offer no real added clinical value over standard stems [102]. In a radiographic comparison between 95 dual modular stems (Profemur Z, Wright Medical Technology Inc.) and 95 match controlled monoblock stems of otherwise similar design (Alloclassic, Zweymüller, Zimmer), the use of dual modular stems revealed no clear benefit in restoring hip geometry and dislocation rate [103]. A cohort study of 324,108 patients found poorer survivorship with interchangeable neck THAs than with monoblock stems independent of other revision risk factors, which implies that patients receiving dual modular implant are not being given the best possible chances compared to patients with fixed-neck THAs [104].

In general, Ti alloy implants have performed better than CoCr alloy implants in the long-term [33,105]. Interchangeable necks made from CoCr alloys are stiffer, produce fewer micromovements [13] and thus promote lesser loss of protective oxide layer and fretting corrosion than Ti alloy necks; however, the added galvanic corrosion negates the beneficial effects of a decrease in micromotion, leading CoCr alloy necks to fracture earlier than Ti alloy necks. CoCr necks are also more heavily related to ALTRs [54] and created debris can be otherwise toxic and carcinogenic [2]. They have not proven to be an effective solution to pre-existing problems, and the long CoCr alloy necks were removed from the inventory of the Profemur series and recalled from the market by the manufacturer (MicroPort Orthopedics Inc., Arlington, TN, USA) in 2015.

In recent years, some studies have reported satisfactory results using dual modular stems in primary THAs [106,107,108]. Specific dual modular stems offer a better reconstruction of hips that deviate from standard anatomy when compared to a monoblock arthroplasty system with different offset options [109]. Since dual modularity in primary THA shows a lack of clinical benefit over the monoblock counterparts and provides no functional advantage, surgeons should restrict usage to a minimum (e.g., in cases of dysplasia or other femoral deformities) [46,99,105,110]. To our knowledge, there have been two similar review studies published recently [32,111]. Lex et al. pointed out that even studies that described medium-term revision rates of primary THAs with a Ti-Ti stem-neck junction as acceptable see them as a sensible therapeutic option only in patients with hip deformities not amenable to correction using standard monoblock stems [32]; however, although sometimes statistically presented as a rare complication, modular neck fracture presents a devastating complication for the patient, surgeon, and manufacturer of the implant [111]. Uniform terminology or a standard classification proposal is mandatory to define the location/area as well as the reason for the implant fracture in a standardised manner, to create more transparency, and to ease the comparison of results from different studies and registers [111].

This review has some drawbacks. We have not searched through the Scopus, Ovid, Ebsco, and Embase databases, so there might be some literature that we have missed, and their findings are not included in the discussion. There are not many studies comparing dual modular implant designs to monoblock implant designs in the same population group. We have also not greatly concentrated on national registries of arthroplasties since they usually do not further specify the type or location of implant fracture, but they are all covered under the overarching term implant fracture or even stem failure. A quantitative analysis of reviewed data offering a meta-analytic perspective has not been performed, which can put our qualitative synthesis under bias.

## 5. Conclusions

Corrosion in a neck-stem junction presents a serious complication that can cause worse treatment outcomes regarding a primary THA. Dual modular stems should be avoided in younger, active, and overweight patients, especially those of the male gender. Large-diameter heads and modular neck and interchangeable head combinations resulting in longer lever arms should be used with extreme caution. All patients with implanted recalled implants and dual modular stems should be carefully assessed. CT scans and MARS MRI should be performed to check for osteolysis and pseudotumors. The usage of CoCr implants can lead to toxicity related to Co and Cr ions and different response mechanisms in the body, thus resulting in ALVAL formation. To not overlook such pathologies in an often-unspecific clinical presentation, serum trace element analysis should be carried out to check for metal ion levels. This includes patients with CoCr alloy necks in combination with Ti alloy stems, which are at higher risk for failure. The interchangeable neck fractures might present the tip of the iceberg of corrosion-related drawbacks of using dual modular stems in primary THAs.

## Figures and Tables

**Figure 1 materials-14-03078-f001:**
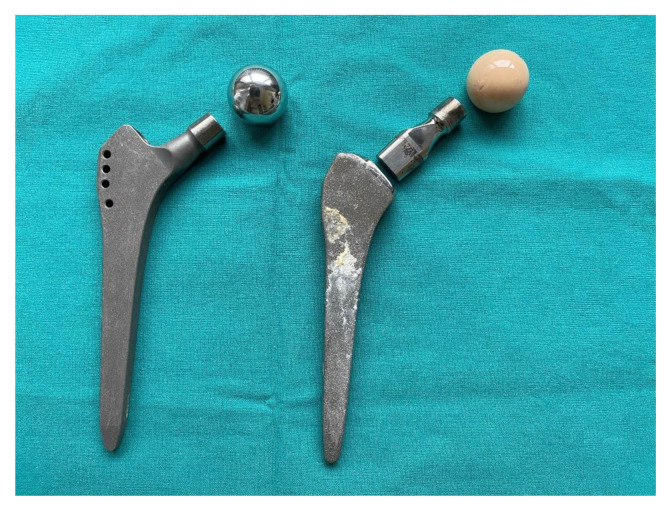
Monoblock (**left**) and dual modular femoral stems (**right**). Both stems could be fitted with separate heads of a different material, size, and bore length. The dual modular stem has been explanted 14 years after primary implantation at revision due to aseptic loosening of the acetabular component and massive proximal femoral bone loss.

**Figure 2 materials-14-03078-f002:**
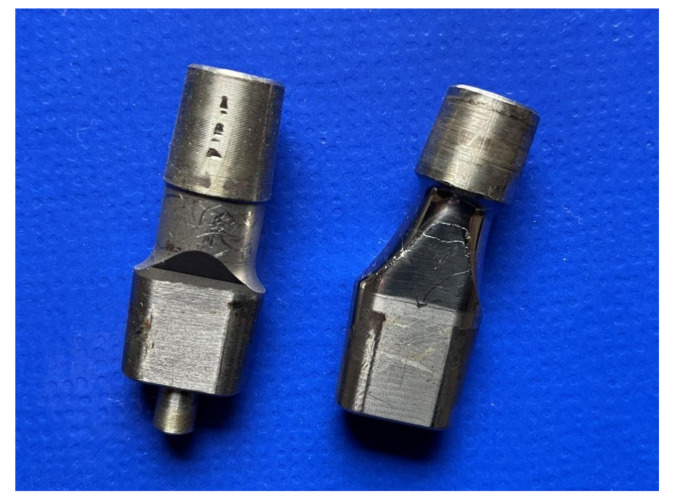
The interchangeable neck of the GSP (Cremascoli S.p.A.) (**left**) and Profemur Z (Wright Medical Technology Inc.) (**right**) dual modular hip implants. Both necks are short, the GSP-neck is straight, and the Profemur Z-neck is in an 8° varus orientation. The GSP neck was explanted 22 years after primary THA because of a late haematogenous infection, and the Profemur Z-neck was explanted 14 years after primary THA due to aseptic loosening. Note the same dimensions of the oval-shaped distal Morse taper.

**Figure 3 materials-14-03078-f003:**
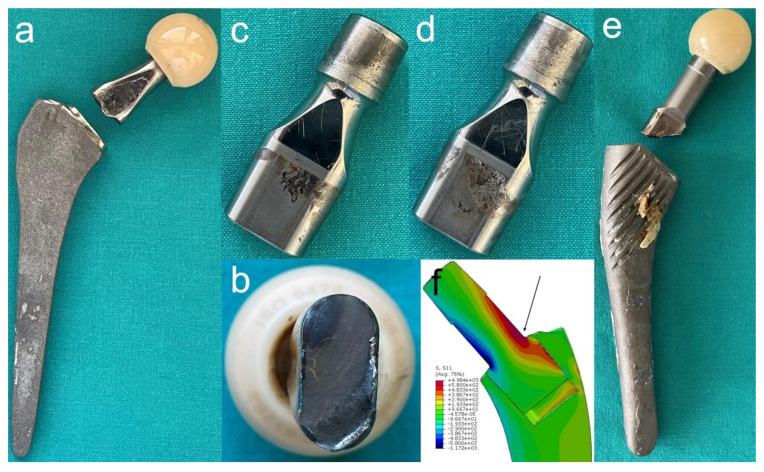
Consequences of corrosion at the interchangeable neck-femoral stem junction. (**a**,**b**) A photograph of a fractured long straight modular neck 15.5 years after implantation of a dual modular femoral stem (Profemur Z, Wright Medical Technology Inc.). Beach marks on the fractured surface are typical for a Ti6Al4V alloy. (**c**,**d**) Both sides of an explanted short straight modular neck 14 years after primary THA. Revision was done due to aseptic loosening of the acetabular component. Note corrosion signs at the oval distal taper. (**e**) A photograph showing a fractured long straight modular neck 20.3 years after implantation of a dual-modular femoral stem (GSP, Cremascoli S.p.A.) The distal part of the neck remained engaged in the stem. (**f**) A 3D model representation of finite element analysis of bending moments acting on the neck-stem junction in an overweight patient with the femoral configuration shown in (**e**) during stumbling, i.e., where peak forces that are eight times the body weight apply. The arrow is showing the site where cracks in the interchangeable necks are usually initiated.

## Data Availability

Data supporting the reported results can be found in the databases mentioned above with the described search methods.

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
