# Peer review of "Dual Modular Titanium Alloy Femoral Stem Failure Mechanisms and Suggested Clinical Approaches"

_materials, 2021, doi:10.3390/ma14113078_

Round 1

Reviewer 1 Report

This review requires revision, and only after that it will be discuss the possibility of its publication.

1) The introduction is very short. There is a lack of critical formulation of the aim and objectives of this review. How exactly can this review help the scientific community?

2) Fig. 1 it is better to transfer to the introduction and it is desirable to show all the implant schemes that exist at the moment.

3) Are there other reviews in the literature on a given topic?

4) Part 3.1. This part of the review needs to be improved. It is necessary to systematize data on the type of implant fractures. It is necessary to make a table: the first column - "type of fracture", the second column - "reasons", the third column - "literature source".

5) Part 4. This part also needs to be finalized. For example, add data on corrosion of the Ti6Al4V alloy in biological media. What is the average lifespan of implants before they break?

6) The conclusions are very small. There is no critical view and recommendations of the authors based on the analysis of the literature data.

Reviewer 2 Report

This is a comprehensive overview of the modular connection problem with modular femoral prostheses. The problem was recognized by orthopedic surgeons years ago, and modular necks are now only used in exceptional cases in primary cases and are more likely to be used in revision arthroplasty. Nevertheless it is still an existing issue in total hip arthroplasty and every surgeon must be aware of this risk.

The issue is well-described and supported by relevant literature.

I have made only a few remarks/comments. They are included in the reviewed manuscript that is attached.

Round 2

Reviewer 1 Report

The authors have completed the review manuscript quite well, but there are several points that, in my opinion, will significantly improve the presented manuscript.

1) Line 103. Table 1 is in the supplementary materials. This should be noted in the text of the manuscript so as not to confuse the reader.

2) Fig. 3 should be expanded (according to different sources) for better visualization of the causes of the destruction of implants. For example, a) and b) corrosion of titanium alloy after n years of working; c) and d) mechanical destruction of the titanium alloy after n years of working; e) and f) destruction of the CoCr alloy.

3) Discussion. Are there data on implants corrosion inhibitors?

4) Discussion and Conclusion. Why do patients need serum analysis for CoCr implants? Describe this more clearly (several sentences), possibly indicating the benefits of titanium alloys for medical applications.

Round 3

Reviewer 1 Report

Well done!